# Chemical Constituents, Anticancer and Anti-Proliferative Potential of *Limonium* Species: A Systematic Review

**DOI:** 10.3390/ph16020293

**Published:** 2023-02-14

**Authors:** Naiara Cássia Gancedo, Raquel Isolani, Natalia Castelhano de Oliveira, Celso Vataru Nakamura, Daniela Cristina de Medeiros Araújo, Andreia Cristina Conegero Sanches, Fernanda Stumpf Tonin, Fernando Fernandez-Llimos, Danielly Chierrito, João Carlos Palazzo de Mello

**Affiliations:** 1Laboratory of Pharmaceutical Biology, Department of Pharmacy, Universidade Estadual de Maringá, Palafito, Maringá 87020-900, Brazil; 2Laboratory of Technological Innovation in the Development of Drugs and Cosmetics, Department of Basic Health Sciences, Universidade Estadual de Maringá, Maringá 87020-900, Brazil; 3Department of Pharmacy, Centro Universitário Ingá, Maringá 87035-510, Brazil; 4Department of Medical and Pharmaceutical Sciences, Universidade Estadual do Oeste do Paraná, Cascavel 85818-760, Brazil; 5Pharmaceutical Sciences Post-Graduate Research Program, Universidade Federal do Paraná, Curitiba 80210-170, Brazil; 6H&TRC—Health & Technology Research Center, ESTeSL—Escola Superior de Tecnologia da Saúde, Instituto Politécnico de Lisboa, 1990-096 Lisboa, Portugal; 7Laboratory of Pharmacology, Department of Drug Sciences, Faculty of Pharmacy, University of Porto, 4050-313 Porto, Portugal

**Keywords:** antitumor activity, cytotoxicity, metastasis, phytochemicals, Plumbaginaceae, qualitative synthesis

## Abstract

*Limonium* species represent a source of bioactive compounds that have been widely used in folk medicine. This study aimed to synthesize the anticancer and anti-proliferative potential of *Limonium* species through a systematic review. Searches were performed in the electronic databases PubMed/MEDLINE, Scopus, and Scielo and via a manual search. *In vivo* or *in vitro* studies that evaluated the anticancer or anti-proliferative effect of at least one *Limonium* species were included. In total, 942 studies were identified, with 33 articles read in full and 17 studies included for qualitative synthesis. Of these, 14 (82.35%) refer to *in vitro* assays, one (5.88%) was *in vivo*, and two (11.76%) were designed as *in vitro* and *in vivo* assays. Different extracts and isolated compounds from *Limonium* species were evaluated through cytotoxic analysis against various cancer cells lines (especially hepatocellular carcinoma—HepG2; n = 7, 41.18%). *Limonium tetragonum* was the most evaluated species. The possible cellular mechanism involved in the anticancer activity of some *Limonium* species included the inhibition of enzymatic activities and expression of matrix metalloproteinases (MMPs), which suggested anti-metastatic effects, anti-melanogenic activity, cell proliferation inhibition pathways, and antioxidant and immunomodulatory effects. The results reinforce the potential of *Limonium* species as a source for the discovery and development of new potential cytotoxic and anticancer agents. However, further studies and improvements in experimental designs are needed to better demonstrate the mechanism of action of all of these compounds.

## 1. Introduction

In the past years, several reviewed articles have summarized the anti-proliferative activity of different phytochemicals, actively contributing to evidence synthesis and knowledge dissemination and of which findings may guide further *in vitro* and *in vivo* studies [1,2,3,4,5,6,7].

According to the literature, nearly 80% of the world’s population depends on traditional medicines to manage a range of diseases, including cancer. Among the clinically approved anticancer drugs, over 50% are derivatives of medicinal plants, as these have been recognized as a source of biologically active compounds with therapeutic potential, being historically used to treat, among others, different types of tumors [8,9,10]. In the United States of America, around 50–60% of oncology patients use agents derived from parts of plants or their nutrients (i.e., complementary and alternative medicines), exclusively or concomitantly with usual therapeutic regimens, such as chemotherapy or radiotherapy. These include curcumin from turmeric, genistein from soybean, polyphenols from green tea, resveratrol from grapes, lycopene from tomato, and gingerol from gingers [1,3].

In this setting, Plumbaginaceae is a family that has 22 accepted genera including *Limonium* Mill., which has 607 accepted species, six of which have been recorded as synonyms [11,12]. Previous phytochemical studies on the *Limonium* species demonstrated the presence of different classes of metabolites, such as anthocyanins, flavonoids, proanthocyanidins, hydrolysable tannins, phytosterol, saponins, phenolic acids, and essential oils [13,14,15]. Additionally, *Limonium* includes species used in folk medicine that have been associated with a range of biological activities, such as antioxidant activities and free radical-scavenging abilities, antibacterial, antifungal, antimalarial, antileishmanial and neuroprotective effects, and promising cytotoxic activity against cancer cells [16,17,18,19,20,21,22,23].

Although some primary studies evaluated the anticancer activity of *Limonium* species, articles that have systematically synthesized all available evidence on the potential roles of this genus in the oncology field are scarce [24]. Thus, this study aimed to assess the anticancer and anti-proliferative potential of *Limonium* species by means of a broad systematic review of the literature.

## 2. Results

### 2.1. Literature Search Results and Main Characteristics of Included Studies

Overall, 942 records were identified in the database after duplicate removal, of which 909 were excluded during screening (based on reading the title and abstract). Of the 33 articles read in full, 16 were eligible for inclusion. One additional record was found during manual searches, resulting in 17 studies for synthesis as shown in Figure 1 [18,22,25,26,27,28,29,30,31,32,33,34,35,36,37,38,39]. See the list of the studies excluded after full-text reading in the Appendix A.

The main baseline characteristics of the included studies are depicted in Table 1. Overall, 14 studies (82.35%) were designed as *in vitro* assessments, one study (5.88%) was *in vivo*, and two (11.76%) evaluated both *in vivo* and *in vitro* parameters. No *ex vivo* and *in silico* studies were found. Studies were mostly performed in China (n = 5, 29.41%) and published between 2003 and 2021.

The main reported human cancer cell lines were acute lymphoblastic leukemia (MOLT-4), acute promyelocytic leukemia (HL-60), breast adenocarcinoma (MCF-7), breast carcinoma (T-47D), cervix adenocarcinoma (HeLa), colorectal adenocarcinoma (DLD-1, HT-29, LoVo), colorectal carcinoma (HCT116), chronic myeloid leukemia (K562), diffuse large cell lymphoma or non-Hodgkin’s B cell (Toledo), fibrosarcoma (HT-1080), gastric adenocarcinoma (BGC-823), hepatocellular carcinoma (HepG2), lung carcinoma (A549), malignant melanoma (SK-MEL-28), osteosarcoma (U2-OS), and pancreas epithelioid carcinoma (PANC-1); meanwhile, non-human cancer cell lines included Abelson murine leukemia virus-induced tumor (RAW-264.7), mouse melanoma (B16-F10), mouse sarcoma (J774), and rat glioma (C6). For human normal cell lines, studies used embryonic kidney (HEK-293), lung fibroblast (WI-38), primary peripheral blood mononuclear cells (PBMC), and skin fibroblast (WS1); for non-human normal cell lines, mouse bone marrow (S17), mouse microglia (N9), and monkey kidney (Vero) were reported.

The most evaluated *in vitro* cell line was HepG2 for human hepatocellular carcinoma (n = 7, 43.75%). Thirteen studies (76.47%) also evaluated other biological activity of *Limonium* species, such as immunologic effects; antioxidant, anti-diabetic, anti-inflammatory, antimicrobial, and antiviral activities; and anti-migration and anti-clonogenic effects in cells.

The aerial parts were the most used, followed by the underground plant organs. Only one study (5.88%) used the whole plant, and other four (23.53%) were unclear about the part of the plant used. Most studies (n = 7; 41.18%) analyzed the *in vivo* or *in vitro* activities using only crude extracts, while the other four (23.53%) evaluated just isolated compounds of *Limonium* species. Both crude extracts and fractions of the plant were analyzed in one study (5.88%), while two others (11.76%) assessed only the fractions. The remaining three articles (17.64%) analyzed crude extracts, fractions, and isolated compounds of the plant. The phytochemistry of *Limonium* species included in the systematic review is summarized in Table 2.

Around one-third of studies (n = 5; 29.41%) assessed the potential mechanism of action of the tested compounds. Bae et al. [29] evaluated the matrix metalloproteinase (MMP) enzymatic activities and expression inhibition with *Limonium tetragonum* (Thunb.) Bullock, and Bae et al. [31] further elaborated on this activity, while Lee et al. [34] tested the anti-melanogenic effects of *L. tetragonum* via tyrosinase and tyrosinase-related proteins. Hamadou et al. [37] assessed the pro-apoptotic property of *Limonium duriusculum* (Girard) Kuntze, and Cordeiro et al. [33] used flow cytometry to evaluate the cell death pathway caused by *Limonium brasiliense* (Boiss.) Kuntze compounds.

Other anti-neoplastic drugs were also used as positive controls (e.g., amsacrine, etoposide, camptothecin, cyclophosphamide, doxorubicin, 5-fluorouracil, kojic acid, lentinan). Three of the sixteen *in vitro* studies (18.75%) did not present data on the cell culture conditions; only one *in vitro* study (6.25%) calculated the selectivity index (SI). The SI can be defined as the ratio of the toxic concentration of a sample against its effective bioactive concentration. For evaluating any anti-proliferative activity of a sample, its cytotoxicity against normal and cancer cell lines must be determined in order to calculate the SI value [40].

The treatment time ranged from 24 to 72 h for *in vitro* assays and from 6 to 312 h for *in vivo* assays. The results based on the 50% inhibitory concentration (IC_50_) of cell proliferation (Table 3) and the main results of *in vivo* and *in vitro* assays of all eligible studies included in this systematic review are summarized in Figure 2.

### 2.2. Anticancer and Anti-Proliferative Activities of Limonium Species

The toxic activity of *L. vulgare* Mill. ethanolic extract was evaluated using larvae and adults of *Artemia salina* and *Daphnia magna*, respectively. Results showed that this species presented maximum toxicity (>50%) against *A. salina* (both larvae and adults), and the chronic toxicity was considered higher than acute toxicity, as set by Lellau and Liebezeit [25]. For *D. magna* adults, *L. vulgare* reached relative toxicity of around 40%. The authors demonstrated that the *L. vulgare* extract was the second lead sample with the maximum activity for the inhibition of tumor induction based on a potato disc assay [25].

The study of Tang et al. [26] showed that polysaccharides of *Limonium sinense* (Girard) Kuntze (LSP) had no significant growth inhibition effect *in vitro* against HeLa and K562 cell lines. Although LSP could inhibit the growth of HepG2 cells, the maximal inhibition rate of LSP was no more than 30% for the concentration tested of 500 µg/mL. On the other hand, for all three different doses of LSP, *in vivo* tests demonstrated an important anticancer activity on Heps tumor cells. The greatest tumor inhibition rates achieved with 400 mg/kg of LSP were 38.03%. The LSP improved macrophage phagocytosis functions in immune-suppressed mice, suggesting that the anticancer activity of this compound can be related to the regulation of immune functions in mice [26]. Another study from this research group using isolated and purified polysaccharides of *L. sinense* (LSP11, LSP21, LSP31) revealed that LSP21 has the most significant dose-dependent inhibitory effect on the growth of HepG2 tumor cells (inhibitory rate of 48.13%). This isolated compound induced cell body shrinkage, chromatin condensation, and a decrease in the number of tumor cells with normal morphology, which suggested that its cytotoxicity can be related to the inhibition of cell proliferation and induction of cell death [28].

A new flavonoid glycoside isolated from *Limonium franchetii* Kuntze **(1)** had moderate *in vitro* cytotoxic activity against the C6 cell line, with a proliferation inhibition rate of 77.09% (100 µg/mL). However, other isolated compounds had no significant cytotoxic activity against BGC-823 and HepG2 cell lines [27]. The first study on the anti-proliferative activity of *Limonium densiflorum* Kuntze, performed by Medini et al. [18], showed dichloromethane extract as having important cytotoxic activity against A549 and DLD-1 cell lines, with IC_50_ values of 29 µg/mL and 85 µg/mL, respectively. Furthermore, this extract was not significantly cytotoxic against the normal human WS1 cell line [18].

Bae et al. [29] suggested that the *L. tetragonum* extract was cytocompatible with the human HT-1080 cell lines and inhibited the enzymatic activity and mRNA expression of matrix metalloproteinase (MMP-2 and MMP-9). In another study, Bae et al. [31] evaluated the anti-metastasis effect of *L. tetragonum* extract against HT-1080 cell lines, focusing on the inhibition of matrix metalloproteinases (MMP-2 and MMP-9) and the regulation of MMPs by intracellular inhibitors called tissue inhibitors of metalloproteinase (TIMPs). The authors demonstrated that 85% methanol and *n*-butanol fractions of the plant had potential antimetastatic effects and can regulate cell proliferation, differentiation, and death through their inhibitory effects on the enzymatic activity of MMPs (MMP-2 and MMP-9), regulation of MMPs and TIMP expression, and suppression of the mitogen-activated protein kinase (MAPK) pathway. However, *n*-hexane and 85% methanol fractions exhibit increased cytotoxicity following high concentrations. All fractions were cytocompatible at concentrations below 50 µg/mL. Similar results were found by Lee et al. [34] who additionally revealed that 85% methanol and *n*-butanol fractions of *L. tetragonum* had antimelanogenic activity due to tyrosinase-inhibitory effects, the prevention of L-3,4-dihydroxyphenylalanine (L-DOPA) oxidation, and suppression of melanin production [34].

The preliminary toxicity screen of an aqueous extract of *Limonium algarvence* Erben flowers against mammalian cell lines (HepG2, N9, and S17) and brine shrimp eggs (*Artemia salina*) was evaluated by Rodrigues et al. [30]. The *in vitro* study resulted in rates of cellular viability higher than 80% at the concentration of 100 µg/mL, and non-toxic effects were observed at the maximal concentration of 1000 µg/mL against *A. salina*. According to the authors, all ethanoic extracts of *L. algarvence* flowers, peduncles, and leaves had no toxicity against human normal and cancer cell lines, HEK 293 and HepG2 cells, respectively. However, few extracts were able to reduce the viability of the non-human cancer cell line (RAW 264.7), with cellular viabilities ranging from 67.4% to 78.2% [38].

Chen et al. [32] evaluated the anti-proliferative activity of isolated compounds of *Limonium bicolor* Kuntze flowers. Both luteolin **(4)** and quercetin **(9)** were cytotoxic against the LoVo cell line, with rates of 89.10% and 79.78% for cell proliferation inhibition, respectively, at 100 µg/mL. The compounds acacetin **(15)** and eriodictyol **(16)** were cytotoxic against the U-2OS cell line at 100 µg/mL, with cell proliferation inhibition of 96.83% and 82.06%, respectively. Only acacetin was able to inhibit the proliferation of the MCF-7 cell line (97.05% at 100 µg/mL and 68.39% at 20 µg/mL). The authors suggested that the presence of 3-*O*-glycosylation in the isolated flavonoid of *L. bicolor* is not paramount for cytotoxic activity [32].

The cytotoxicity of crude extracts, fractions, subfractions, and isolated compounds (epigallocatechin-3-*O*-galatte **(28)**, samaragenin A **(29)**, and samaragenin B **(30)**) of *L. brasiliense* rhizome was evaluated by Cordeiro [33]. The values of the SI of aqueous and ethyl-acetate fractions corresponded to a selectivity four times higher for neoplastic cells (HL-60 cell line) compared to that for normal cells (PBMC cell line). The most promising anti-neoplastic activity was against human acute promyelocytic leukemia cells (HL-60) with the subfractions F and G (IC_50_ = 8.23 ± 0.83; IC_50_ = 7.35 ± 0.36 µg/mL, respectively). The subfraction G showed an IC_50_ value of 7.92 ± 0.86 µg/mL against the MOLT-4 cell line, while samaragenin A resulted in an IC_50_ value of 29.24 ± 17.64 µg/mL for the K562 cell line. According to flow cytometry results, subfraction G required the lowest concentration for cell death mediated by apoptosis induction for K562 cell line (10 µg/mL) and the highest percentage of cell death mediated by late apoptosis (37.8%) and necrosis (24.7%) at 50 µg/mL for the MOLT-4 cell line, after 48 h of treatment. The isolated compound, samarangenin A, did not cause significant cell death (*p* < 0.05) [33].

Sahli et al. [35] evaluated the cytotoxic activity of a methanol crude extract of stems and leaves of *Limonium virgatum* (Willd.) Fourr. The crude extracts were more cytotoxic against the non-human tumor cell line (J774) than against the human non-tumor cell line (WI-38). However, the extracts of the species *Silene succulenta* Forssk. (stem and leaves) and *Cirsium scabrum* (Poir.) Bonnet & Barratte (leaves) showed the most significant cytotoxic activities when compared with those of *L. virgatum* [35].

According to Al-Madhagi et al. [22], a petroleum ether extract of *Limonium sokotranum* (Vierh.) Radcl.-Sm. leaves and flowers exhibited the highest cytotoxic activity against HepG2 tumor cells, with an IC_50_ value of 9.97 ± 0.79 μg/mL, which was close to that of the positive control, doxorubicin (7.38 ± 0.11 μg/mL). On the other hand, IC_50_ values in the test against the MCF-7 cell line ranged from 8.70 ± 0.08 to 21.8 ± 1.30 μg/mL, and the lowest IC_50_ value was recorded with the methanol extract of *L. sokotranum* leaves and flowers (8.70 ± 0.08 μg/mL) [22].

The anti-proliferative activity of an *n*-butanol extract from aerial parts of *Limonium bonduellei* (T.Lestib.) Kuntze against two human cancer cell lines (HT-29 and HeLa) was evaluated by Amrani et al. [36]. The extract showed a concentration-dependent anti-proliferative effect. Low concentrations showed better activity against the HeLa cell line at 15 h and HT-29 cell line at 30 h, after treatment. The highest concentration of an *n*-butanol extract of *L. bonduellei* (250 μg/mL) showed the highest proliferation inhibition in all cell lines (92.6% in HT-29 and 98.9% in HeLa) [36].

The anti-proliferative and pro-apoptotic activities of an *n*-butanol extract and isolated compounds (apigenin **(31)** and apigenin7-*O*-β-D-(6”-methylglucuronide) **(32)**) of *L. duriusculum* against the HCT116 cell line were assessed for the first time by Hamadou et al. [37]. The authors showed that the crude extract had an IC_50_ value of 7.60 μg/mL, while the results for the apigenin IC_50_ were 25.74 μM. Apigenin7-*O*-β-D-6”-methylglucuronide did not affect cell proliferation [37].

The cytotoxic activity of the ethyl-acetate extract and isolated lignanamides of *Limonium gmelinii* Kuntze roots against tumor cell lines was evaluated by Tuohongerbieke et al. [39]. The ethyl-acetate extract showed moderate cytotoxicity against the HeLa cell line (IC_50_= 25.25 μg/mL), and compounds **(33)** (IC_50_ = 19.24 ± 1.62 μM) and **(50)** (IC_50_= 12.85 ± 2.65 μM) and compounds **(37)**, **(43)**, **(33)**, and **(50)** demonstrated moderate cytotoxicity against the MCF-7 cell line, with IC_50_ values ranged from 14.14 ± 1.08 to 28.85 ± 2.33 μM. Other lignanamides showed low or no cytotoxicity (IC_50_ >30 μM).

According to SYRCLE’s tool, some *in vitro* studies did not properly describe the conditions of cell culture (n = 3, 18.75%) in the Appendix A. All *in vitro* and *in vivo* studies were unclear about the domain of baseline characteristics, allocation concealment, and incomplete outcome data. It is unclear whether both *in vitro* and *in vivo* studies were free of selective data reporting, especially due to the lack of conflicts of interest or funding statements in some articles. Two-thirds of the articles were unclear about other sources of bias (see Appendix A).

## 3. Discussion

This is the first systematic review to gather evidence on the anti-proliferative and anticancer activities of *Limonium* species. This genus includes one of the most interesting halophyte plants that grow under several abiotic stress conditions, and it is responsible for providing molecules with important bioactive properties [23,41,42].

However, despite *Limonium* species having been widely used in folk medicine, there are few studies about the biological potential of this genus, as observed by Medini et al. [18] and during our literature research. This study included 17 studies, most of which were designed as *in vitro* assays evaluating the cytotoxicity of different extracts, fractions, subfractions, and isolated compounds of the *Limonium* species using a range of cell cultures (both human and non-human cancer cell lines, as well as human and non-human normal cells).

One of the aims of the *in vivo* and *in vitro* screening of natural products is to discover new promising agents, such as those with anticancer activity, and guide the development of new drugs [43]. According to Kuete and Efferth [44] for *in vitro* anticancer screenings of plant extracts, we can consider significant or strong cytotoxicity values of IC_50_ below 20 µg/mL for extracts and below 10 µM for isolated compounds. Usually, anticancer drugs from natural compounds act by inhibiting DNA synthesis (antimetabolites), damaging DNA (DNA alkylating agents and topoisomerase poisons), or inhibiting the function of the mitotic spindle based on microtubes (e.g., taxanes) [45,46,47]. In the studies included in this review, the antineoplastic drugs used as positive controls act by inhibiting the enzymes DNA topoisomerase I (camptothecin) and DNA topoisomerase II (amsacrine, doxorubicin, etoposide); damaging DNA as alkylating agents (cyclophosphamide), inhibiting the synthesis of pyrimidine, and thus the formation of DNA (5-fluorouracil); modulating the immune system (lentian); and as melanogenesis inhibitors with potent tyrosinase-inhibitory activity (kojic acid) [48,49,50].

Most *in vitro* studies were based on the colorimetric assay 3-(4,5-dimethylthiazol-2-yl)-2–5-diphenyltetrazolium bromide (MTT), a cell proliferation assay that measures the activity of mitochondrial dehydrogenase enzymes in living cells, and it is one of the most widely used assays for evaluating the preliminary anticancer activity of natural products [51,52]. Other studies used the colorimetric assay sulforhodamine B or fluorometric assays as resazurin reduction, Hoechst 33342, and calcein-acetomethoxy (Calcein-AM). *In vitro* cell viability and cytotoxicity assays using cultured cells are widely employed for drug screening and have some advantages, such as quick assays, reduced costs, and room for automation. Currently, these assays are also used in anticancer drug development to evaluate the cytotoxicity and tumor cell growth inhibition of different compounds. However, *in vitro* assays are not technically advanced enough to promptly replace animal tests due to the lack of a physiological environment. This is not the case of *in vivo* assays that are able to measure several behavioral and physiological parameters and guide the understanding of the pharmacological activity of the tested compound on the entire organism [53,54].

The toxicity of *Limonium* species was evaluated through bioassays using the organisms *A. salina* and *D. magna*. *A. salina* (brine shrimp) is a highly sensitive crustacean, and it has been extensively used for toxic screening of bioactive compounds since 1956 [55]. *D. magna* was first mentioned by Flücker and Flück (1949) as another organism used in toxicity testing. It is a simple, sensitive, and reproducible laboratory model for the toxicity screening of compounds [56,57,58,59]. These bioassays are well correlated with cytotoxicity and are used to screen the potential anti-tumor activity of natural compounds [60]. The ethanolic extract of *L. vulgare* was toxic against *A. salina* and can be considered a promising candidate for new anticancer compounds [25]. The aqueous extract of *L. algarvense* was non-toxic against *A. salina*, which could explain the lack of toxicity against the human hepatocellular carcinoma cell line [30]. This suggests a good correlation of preliminary toxicological evaluations of plant-derived compounds using *in vitro* mammalian cells and *in vivo* brine shrimp assays.

On the other hand, few ethanolic extracts of different organs of *L. algarvense* have low cytotoxicity against the RAW 264.7 mouse cell line, and all extracts were not toxic against the human normal HEK 293 cell line and the human tumor HepG2 cell line. The authors suggest a possible correlation between the presence of several flavonoids in *L. algarvense* and the *in vitro* and *in vivo* hepatoprotective effect as demonstrated by other literature studies with this class of secondary metabolites [38]. Despite the low anti-proliferative activity of the crude polysaccharides of *L. sinense* against HepG2 tumor cells, this compound inhibited the growth of transplanted mouse tumors and demonstrated a synergistic action when used in association with the anti-neoplastic agent 5-fluorouracil. It was suggested that the anti-cancer effects could be related to the *in vivo* immunomodulatory activity [26].

*L. tetragonum*, the most evaluated species among the included studies, was shown to be a potential source of bioactive agents with proven anti-MMP activity and anti-melanogenesis properties, including compounds that can prevent hyperpigmentation [29,31,34]. Bae et al. [31] and Lee et al. [34] suggest that the active compounds of *L. tetragonum* include, but are not limited to, flavonoid glycosides (e.g., myricetin 3-galactoside **(13)** and quercetin 3-*O*-β-galactopyranoside **(14)**). These compounds inhibit the activity of MMP, suppress MAPK associated with MMP upregulation, and act as anti-melanogenic compounds, demonstrating the nutraceutical potential of *L. tetragonum* as a source of anti-MMP compounds [29,31,34]. The dichloromethane extract of *L. densiflorum* demonstrated promising anti-proliferative effects against human lung carcinoma (A549) and human colorectal adenocarcinoma (DLD-1), with results that were similar to those from the positive control etoposide. Furthermore, this extract was not significantly cytotoxic against a human normal cell line (WS1), which suggests the possible selectivity of the extract for cancer cells [18].

The antioxidant, total phenolic content, and anti-inflammatory activity of *L. densiflorum* crude extracts and isolated compounds was investigated. All extracts reduced nitric oxide (NO) production in a concentration-dependent manner, suggesting interesting anti-oxidant activities. These results can be due, in part, to the majority presence of polyphenolic compounds (flavonoids and phenolic acids) that can be related to the anticancer potential of *L. densiflorum* [18]. In addition, evidence suggests that natural antioxidants are able to inhibit oxidative stress and restore cellular homeostasis, preventing damage to normal tissues and inflammation, which can be valuable for the management of different chronic and metabolic conditions, such as cancer [61,62].

Cordeiro [33] demonstrated a greater SI of aqueous and ethyl-acetate fractions of *L. brasiliense* against neoplastic cells (HL-60) vs. normal cells (PBMCs), and the favorable anti-proliferative activity of subfractions F and G against the HL-60 cell line, which can be related to the immunomodulatory activity of crude extracts and fractions of *L. brasiliense*. In the literature, it is suggested that tumor growth and proliferation can also be restrained by targeting and modulating the immune response. Natural immunomodulators can stimulate humoral and cell-mediated immune responses against the tumor [63,64].

Other important cytotoxic activity, against HepG2 tumor cells, was obtained with the petroleum ether extract of *L. sokotranum* leaves and flowers, which displayed a profile similar to that of the positive control doxorubicin [22]. A methanol extract of this species (leaves and flowers) had the lowest IC_50_ value against the MCF-7 tumor cells (8.70 ± 0.08 µg/mL). Finally, the isolated compound apigenin **(31)** of *L. duriusculum* had the lowest IC_50_ value against the HCT116 cell line (25.74 µM). The *n*-butanol extract and isolated compound apigenin promote apoptosis in HCT116 cancer cells, associated with reduced signaling from MAPK, activation of the p53 response pathway, and poly(ADP-ribose) polymerase (PARP) cleavage [37]. The research article produced by Tuohongerbieke et al. [39] is the first report of lignanamides in Plumbaginaceae. The ethyl-acetate extract and isolated lignanamides from *L. gmelinii* roots showed moderate cytotoxicity against the HeLa cell line (25.25 µg/mL). The possible anti-cancer mechanisms of *Limonium* species suggested in the included studies are summarized in Figure 3.

Many isolated or identified compounds, as primary and secondary metabolites, and principally phenolic compounds were described in the literature sourced. The extraction method and solvent polarity were some factors related to the phytochemical diversity observed in the crude extracts, subfractions, fractions, and isolated compounds of *Limonium* species. Several of the compounds have already been described in *Limonium* spp., such as flavonoids and their glycosides derivatives [32,37,38]. Other compounds were discovered from this genus for the first time (e.g., some lignanamides) [39]. Thus, we suggest that *Limonium* species could be investigated as a source of bioactive phytochemicals, including polyphenolic compounds that might combat oxidative stress, act in cell cycle regulation, and could possibly be used in the nutraceutical field. This can be supported by the fact that *Limonium* spp. are mainly composed of flavonoids, phenolic acids, and tannins (Table 3). These phenolic compounds of *Limonium* spp. can act in the protection of oxidative and inflammatory-related diseases, as suggested by Rodrigues et al. [35] in a comparison study between *L. algarvense* and *Camellia sinensis* (L.) Kuntze (green tea). The authors demonstrated that *L. algarvense* flowers had similar or higher *in vitro* antioxidant and anti-inflammatory properties than green tea, based on radical-scavenging activities and the decrease in NO production, respectively [30].

In addition, several studies demonstrated the antioxidant potential of *Limonium* species associated with their high polyphenol content [18,36,65]. Overall, studies suggest that oxidative stress, chronic inflammation, and cancer are closely related [66]. Thus, the antioxidant activity, as well as anti-inflammatory and immunomodulatory effects of *Limonium* species reported in included studies, can contribute to the anticancer effect observed in some of these species. Furthermore, as *Limonium* species are considered halophytes, which means that they can adapt to salinity conditions via physiological and biochemical processes, a consequent increase in the enzyme and antioxidant metabolites may occur depending on the environment [41,67,68]. Based on these correlations, the literature suggested that phenolic compounds with potent antioxidant activity could be evaluated as possible chemo-preventive or chemotherapeutic agents [69].

The assessment of cytotoxicity and other biological activities from plant species, both *in vitro* and *in vivo* studies, should be performed using appropriate and validated methods aimed at obtaining accurate and reliable results that can be reproduced by other studies. However, some of the included studies were unclear or lacked in their reporting of relevant information, such as the correct and accepted species name (e.g., *L. sokotranum* instead of *L. socotranum*; *L. bonduellei* instead of *L. bonduelli*; *L. franchetii* Kuntze instead of *L. franchetii*; *L. densiflorum* Kuntze instead of *L. densiflorum*; *L. bicolor* Kuntze instead of *L. bicolor*), the organ of the plant material used for extraction, the number of plant voucher specimens, culture conditions of cell lines, and the use of positive and negative controls. It was also found that in SYRCLE’s tool, the general quality of the articles was moderate. In this scenario, it was suggested that a checklist grounded on pharmacognostic literature of medicinal plants should be completed by researchers and authors prior to publication to standardize the conduction and reporting of studies in this field (see Appendix A). It was also encouraged that another checklist published by Chierrito et al. [70] be used for reporting experimental *in vitro* studies, including data on cell culture (e.g., identification of culture type, growth medium used, number of passages, incubation temperature (exact 0.0 °C), atmosphere conditions (exact 0.0% CO_2_), and methods used).

This conducted systematic review has some limitations. Although there is extensive literature on *Limonium* species, only a few studies were included because this field of the anti-proliferative and anticancer effects of this genus is still recent (studies published between 2003 and 2021). Despite the popular use of *Limonium*, few species had their biological potential evaluated. Due to the nature of the data, the differences among *Limonium* species, cell lines, and compounds evaluated, quantitative analyses were not possible.

*Limonium* species have a wide worldwide distribution, with vast chemical and biological potential, and are popularly used in several countries. The results obtained from this systematic review reinforce these data and bring a new perspective in the search for useful anticancer agents from natural sources, mainly polyphenols. In this context, it is necessary to carry out more *in vitro* studies for a better understanding of the mechanism of action of these compounds and in this way, direct future *in vivo* and clinical studies, reinforcing the use of natural products in the discovery of less toxic, more selective, and effective phytochemicals for the treatment of different types of cancer.

## 4. Materials and Methods

### 4.1. Study Design

This systematic review was performed following the recommendations of the Preferred Reporting Items for Systematic Reviews and Meta-Analyses (PRISMA) statement (Appendix A), Cochrane Handbook for Systematic Reviews of Interventions, and The Joana Briggs Institute (JBI) [40,71,72]. All the steps (i.e., article screening, full-text reading, data extraction, and methodological quality assessment) were conducted by two reviewers, independently. A third reviewer was consulted in the case of discrepancies. This study was registered in the Open Science Framework (OSF) with the registration DOI 10.17605/OSF.IO/WHBNE.

### 4.2. Systematic Literature Search and Eligibility Criteria

A systematic search was performed based on the electronic databases PubMed/MEDLINE, Scopus, and Scielo with no time or language restrictions (updated on 24 June 2021). The full search strategy is available in the Appendix A. A manual search was also conducted for the reference list of the included articles, in Clinical Trials.gov, and the Brazilian catalog of thesis from the *Coordenação de Aperfeiçoamento de Pessoal de Nível Superior* (CAPES).

We included *in vitro*, *in vivo*, *ex vivo*, or *in silico* experimental studies that evaluated the anticancer or anti-proliferative activity of *Limonium* species (e.g., crude extract, fraction, subfraction, or isolated compounds). Studies evaluating other biological activities of *Limonium* species and with other study designs (e.g., phytochemistry, agronomic perspective, botany, salinity, or cultivation studies) and those published in non-Roman characters were excluded.

### 4.3. Data Extraction and Reporting Evaluation

A standardized form was used to collect data on the studies’ general characteristics (e.g., authors, publication date, country), bioassay type, methodological aspects, and main results. The adapted tool for *in vitro* assays and the Systematic Review Center for Laboratory Animal Experimentation (SYRCLE) tool were used to assess the studies’ risk of bias and methodological quality in the Appendix A [70,73].

## 5. Conclusions

The literature on the potential anticancer effects of *Limonium* species mostly refers to preliminary assessments of the cytotoxicity of different compounds obtained from crude extracts, fractions, subfractions, and isolates and their impact on the viability of a range of cancer cell lines. *Limonium tetragonum* was the most evaluated species, with promising *in vitro* anti-melanogenesis effects. Isolated compounds of the flavonoid class, such as apigenin of *L. duriusculum*, also demonstrated a favorable cytotoxic effect against colorectal cancer, as well as lignanamides of *L. gmelinii* against cervix and breast adenocarcinoma cell lines. However, the complete mechanism of action of all isolated compounds and their effect in *in vivo* models remain unclear for most species.

These findings reinforce the biological potential of *Limonium* spp. as a source for the discovery and development of new potential cytotoxic phytochemicals. However, better planning, experimental designs, and reporting of the results will make future studies more robust and provide better proof to demonstrate the mechanism of action of these compounds.

## Figures and Tables

**Figure 1 pharmaceuticals-16-00293-f001:**
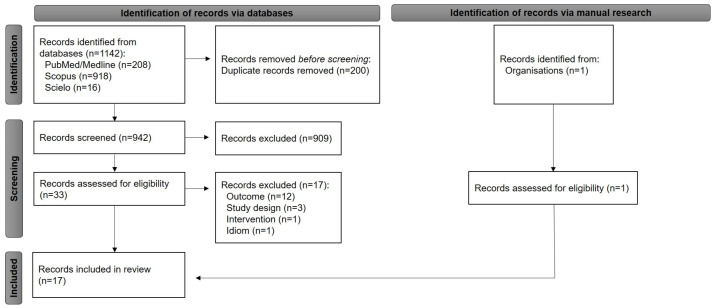
Flow diagram of the systematic review. Fonte: Adapted from Page et al. [40].

**Figure 2 pharmaceuticals-16-00293-f002:**
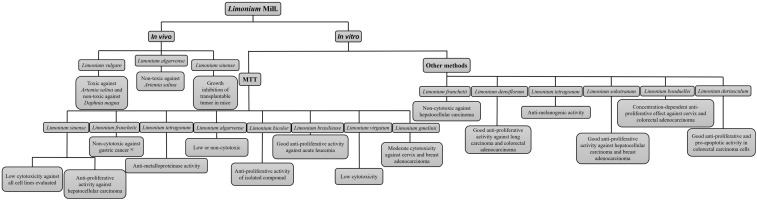
Main results of *in vivo* and *in vitro* assays of eligible studies included in the systematic review. Fonte: Adapted from Medini et al. [18], Al-madhagi et al. [22], Lellau et al. [25], Tang et al. [26,28], Kong et al. [27], Bae et al. [29,31], Chen et al. [32], Cordeiro [33], Lee et al. [34], Sahli et al. [35], Amrani et al. [36], Hamadou et al. [37], Rodrigues et al. [30,38], Tuohongerbieke et al. [39].

**Figure 3 pharmaceuticals-16-00293-f003:**
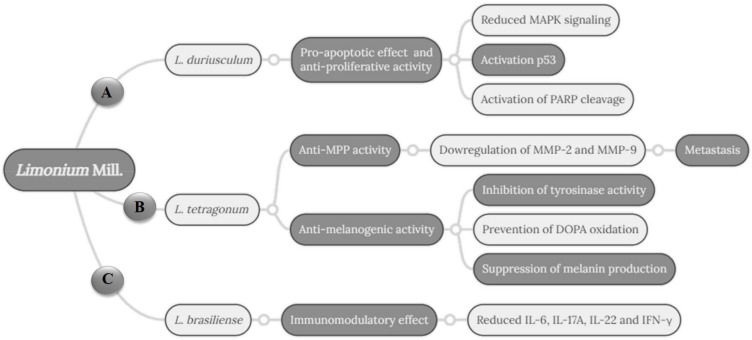
The possible cellular mechanism involved in the anti-cancer activity of *Limonium* species included in the systematic review. (**A**) Cell proliferation inhibition pathways mediated by *L. duriusculum* extract. (**B**) Inhibition of matrix metalloproteinases (MMP-2 and 9), Zn dependent endoproteases related to various complications in cancer, such as metastasis, and anti-melanogenic activity linked to inhibition of melanin biosynthesis mediated by bioactive compounds of *L. tetragonum*. (**C**) Immunomodulatory activity related to reduced of interleukin (IL) 6, 17A, and 22, and interferon-gamma (IFN-ϒ) mediated by *L. brasiliense* crude extracts and fractions, suggesting an anticancer effect of this species. Note: Adapted from Hamadou et al. [37]; Bae et al. [31]; Lee et al. [34]; Cordeiro [33].

**Table 1 pharmaceuticals-16-00293-t001:** Main characteristics of all eligible *in vivo* and *in vitro* studies included in the systematic review.

Reference Number	Country	Plant Species	Part of Plant Used	Cell Line	Bioassay/ Model Used	Compound Tested	Positive Control	Time of Treatment (h)	Other Biological Activities
** *In vivo* **									
**[25]**	Germany	*L. vulgare*	NR	NR	*Artemia salina Daphnia magna*	EtOH extract	HgCl_2_ solution (1%)	6 24 48	NR
**[26]**	China	*L. sinense*	Roots	HepG2	Mice	Crude LSP	Cyclophosphamide Lentinan with 5-fluoracil 5-Fluoracil	312	Immunomodulatory effects
**[30]**	Portugal	*L. algarvense*	Flowers	NR	*Artemia salina*	Aq extract	NR	48	Antioxidant and anti-inflammatory activities
** *In vitro* **									
**[26]**	China	*L. sinense*	Roots	HeLa HepG2 K562	MTT	Crude LSP	NR	24	Immunomodulatory effects
**[28]**	China	*L. sinense*	Roots	HepG2	MTT	LSP11 LSP21 LSP31	5-Fluorouracil	24	NR
**[27]**	China	*L. franchetii*	Whole	BGC-823	MTT	12 isolated compounds	NR	NR	NR
				C6 HepG2	Sulforhodamine B				
**[18]**	Tunisia	*L. densiflorum*	Leaves	A549 DLD-1 WS1	Resazurin reduction test	DCM extract EtOH extract MeOH extract Hex extract	Etoposide	48	Antioxidant and anti-inflammatory activities
**[29]**	Korea	*L. tetragonum*	NR	HT-1080	MTT	DCM fraction	NR	48	Determination of enzymatic activities of MMPs, mRNA expression of MMPs and TIMPs via RT-PCR, and detection of immunoreactive proteins via Western blotting
**[31]**	Korea	*L. tetragonum*	NR	HT-1080	MTT	DCM extract (Hex fraction and 85% MeOH fraction) Aq extract (BuOH fraction and Aq fraction)	NR	48	Determination of enzymatic activities of MMPs, mRNA expression of MMPs and TIMPs via RT-PCR, and detection of immunoreactive proteins via Western blotting
**[34]**	Korea	*L. tetragonum*	NR	B16-F10	Spectrophotometric method	Hex fraction 85% MeOH fraction BuOH fraction Aq fraction	Kojic acid	0.5	DOPA oxidase activity, cellular tyrosinase activity, melanin content, melanogenesis-related mRNA expression via RT-PCR, and detection of TRP via Western blotting
**[30]**	Portugal	*L. algarvense*	Flowers	HepG2 N9 S17 RAW-264.7	MTT	Aq extract	NR	72	Antioxidant and anti-inflammatory activities
**[38]**	Portugal	*L. algarvense*	Flowers Leaves Peduncles	HEK-293 HepG2 RAW-264.7	MTT	EtOH extract	NR	72	Antioxidant activity
**[32]**	China	*L. bicolor*	Flowers	LoVo MCF-7 U-2OS	MTT	15 isolated compounds	5-Fluorouracil	48	NR
**[33]**	Brazil	*L. brasiliense*	Rhizome	HepG2 HL-60 K562 MOLT-4 PANC-1 PBMC SK-MEL-28 T-47D Toledo Vero	MTT	CE Aq fraction EAF Subfractions (A-K) Isolated compounds (SA, SB, EGCG)	Amsacrine	72	Selectivity index, anti-migration and anti-clonogenic potential, and immunomodulatory activity
**[35]**	France	*L. virgatum*	Leaves Stems	J774 WI-38	MTT	MeOH extract	Camptothecin	72	Antiradical, antimicrobial, and antiviral activity
**[22]**	Yemen	*L. sokotranum*	Flowers Leaves Stem	HepG2 MCF-7	Sulforhodamine B	PE extract DCM extract MeOH extract	Doxorubicin	48	Antibacterial and antifungal activity
**[36]**	Algeria	*L. bonduellei*	Flowers Leaves	HeLa HT-29	xCELLigence RTCA	BuOH extract	NR	48 72	DNA damage inhibition efficiency
**[37]**	Algeria	*L. duriusculum*	Flowers Leaves	HCT116	Calcein-AM Hoechst 33342	BuOH extract Apigenin	NR	48	Measures of relative levels of p53, MDM2, p21, total and p-ERK proteins, and PARP cleavage via western blotting
**[39]**	China	*L. gmelinii*	Roots	A549 HeLa MCF-7	MTT	EtOAc extract 19 isolated compounds	Doxorubicin	48	Anti-diabetic and anti-inflammatory activities

Abbreviations: A549: human lung carcinoma; Aq: aqueous; B16-F10: melanoma (mouse); BGC-823: human gastric adenocarcinoma; BuOH: *n*-butanol; C6: brain glioma (rat); Calcein-AM: calcein-acetomethoxy; CE: crude extract; DCM: dicloromethane; DLD-1: human colorectal adenocarcinoma; EAF: ethyl-acetate fraction; EGCG: epigallocatequin-3-*O*-gallate; EtOAc: ethyl-acetate; EtOH: ethanol; HCT116: human colorectal carcinoma; HeLa: human cervix adenocarcinoma; HepG2: human hepatocellular carcinoma; Hex: *n*-hexane; HL-60: human acute promyelocytic leukemia; HT-29: human colorectal adenocarcinoma; HT-1080: human fibrosarcoma; HEK-293: human embryonic kidney (normal cell); J774: sarcoma (mice); K562: human chronic myelogenous leukemia; LoVo: human colorectal adenocarcinoma; LSP: *Limonium sinense* polysaccharide; MMPs: matrix metalloproteinases; MCF-7: human breast adenocarcinoma; MeOH: methanol; MOLT-4: human acute lymphoblastic leukemia; N9: microglia (mice normal cell); NR: not related; PANC-1: human pancreas epithelioid carcinoma; PBMCs: human primary peripheral blood mononuclear cells (normal cells); PE: petroleum ether; p-ERK: phosphorylated ERK; RAW-264.7: Abelson murine leukemia virus-induced tumor (mouse); S17: bone marrow (mouse normal cell); SA: samarangenin A; SB: samarangenin B; SK-MEL-28: human malignant melanoma; T-47D: human breast carcinoma; TIMPs: tissue inhibitor of metalloproteinases; Toledo: human diffuse large cell lymphoma (non-Hodgkin’s B cell); TRP: tyrosinase-related proteins; U-2OS: human osteosarcoma; Vero: kidney (monkey normal cell); xCELLigence RTCA: xCELLigence real-time cell analyzes; WI-38: human lung fibroblast (normal cell); WS1: human skin fibroblast (normal cell). Note: The studies were described in chronological order. The same species were described together.

**Table 2 pharmaceuticals-16-00293-t002:** Phytochemistry of *Limonium* species included in the systematic review.

Reference Number	Plant Species	Class of Metabolite	Compounds	Number of Isolated Compounds Tested *In Vitro **
Primary metabolites				
[28]	*L. sinense*	Polysaccharide	LSP21 (glucose, galactose and mannose)	
[38]	*L. algarvense*	Amino acid	*N*-acetyl-tryptophan	
		Fatty acids	Oxo-tridecanoic acid sulphate Trihydroxy-10-octadecenoic acid Trihydroxy-10,15-octadecadienoic acid	
		Polysaccharide	Hex-3-en-olxylopyranosyl-(1-6)-glicopyranoside Sucrose or isomeric structures	
Secondary metabolites				
[27]	*L. franchetii*	Flavonoids	Apigenin Dihydrokaempferol Kaempferol-3-*O*-α-L-rhamnopyranoside Luteolin Myricetin Myricetin-3-*O*-(2″-*O*-galloyl)-α-L-rhamnopyranoside Myricetin-3-*O*-(3″-*O*-galloyl)-α-L-rhamnopyranoside Myricetin-3-*O*-α-L-rhamnopyranoside Quercetin Quercetin-3-*O*-(2″-*O*-tigloyl)-α-L-rhamnopyranoside Quercetin-3-*O*-(3″-*O*-tigloyl)-α-L-rhamnopyranoside Quercetin-3-*O*- α-L-rhamnopyranoside	**(1)** **(2)** (**3) ** **(4)** **(5)** **(6)** **(7)** **(8)** **(9)** **(10)** **(11)** **(12)**
[18]	*L. densiflorum*	Flavonoids	Catechin hydrate Isorhamnetin Myricetin	
		Phenolic acids	Ellagic acid Gallic acid Sinapic acid *trans* 3-hydroxycinnamic acid	
[30]	*L. algarvense*	Flavonoid	Apigenin	
		Phenolic acids	Caffeic acid Coumaric acid Ferulic acid Gallic acid *p*-Hydroxybenzoic acid Salicylic acid Syringic acid	
[38]		Lignin	Pinoresinol sulphate	
		Flavonoids	2′-C-methyl myricetin-3-*O*-rhamnoside-galloyl 4′-methyl eriodictyol-galloyl-rhamnose Apigenin Apigenin derivative Apigenin-*O*-glucoside Apigenin-*O*-glucuronide Dihydrokaempferol Epigallocatechin gallate Eriodictyol Eriodyctiol-*O*-glucoside Isorhamnetin-3-*O*-rutinoside Licoagroside B Luteolin Luteolin-7-*O*-glucoside Luteolin-7-*O*-rhamnoside Methyl licoagroside B Myricetin Myricetin-3-*O*-(2″-*O*-galloyl)-glucoside Myricetin-3-*O*-acetyl-deoxyhexose Myricetin-3-*O*-acetyl-hexoside Myricetin-3-*O*-pentoside Myricetin-ethyl acetoacetate-galloyl Myricetin-galloyl-acetyl deoxyhexose Myricetin-*O*-(galloyl)-deoxyhexose Myricitin-3-*O*-glucoside Myricitin-3-*O*-rhamnose Myricitin-3-*O*-rutinoside Naringenin Naringenin derivative Quercetin Quercetin derivative Quercetin-3-*O*-rhamnoside Quercetin-hexoside derivative Quercetin-*O*-galloy-glucoside Quercetin-*O*-hexoside Quercetin-tetramethyl ether- -dihydroxyethylfructopyranose Rutin	
		Phenolic acids	Feruloyltyramine Glucosyringic acid Syringic acid	
		Tannins	Digalloyl-hexoside Galloylglucoside derivative Galloyl-hexoside Galloylhexoside derivative	
		Phenylpropanoid	Sinapyl alcohol sulphate	
[31]	*L. tetragonum*	Flavonoids	Myricetin 3-galactoside Quercetin 3-*O*-β-galactopyranoside	**(13)** **(14)**
[34]				
[32]	*L. bicolor*	Flavonoids	Acacetin Eriodictyol Hesperidin Isorhamnetin Kaempferol Kaempferol-3-*O*-(6″-*O*-galloyl)-β-D-glucoside Kaempferol-3-*O*-α-L-rhamnoside Kaempferol-3-*O*-β-D-glucoside Luteolin Myricetin-3-*O*-α-L-rhamnoside Quercetin Quercetin-3-*O*-α-L-rhamnoside Quercetin-3-*O*-β-D-galactoside Quercetin-3-*O*-β-D-glucoside Rutin	**(15)** **(16)** **(17)** **(18)** **(19)** **(20)** **(21)** **(22)** **(4)** **(23)** **(9)** **(24)** **(25)** **(26) ** **(27)**
[33]	*L. brasiliense*	Tannins	Epigallocatequin-3-*O*-gallate Samarangenin A Samarangenin B	**(28)** **(29)** **(30)**
[35]	*L. virgatum*	Phenolic amide	*N*-*trans*-feruloyl tyramine	
[37]	*L. duriusculum*	Flavonoids	Apigenin Apigenin 7-*O*-β-D-(6”-methylglucuronide)	**(31)** **(32)**
[39]	*L. gmelinii*	Lignanamides	(2,3-*trans*)-3-(3-hydroxy-5-methoxyphenyl)-*N*-(4-hydroxyphenethyl)-7-{(*E*)-3-[(4-hydroxyphenethyl)amino]-3-oxoprop-1-en-1-yl}-2,3-dihydrobenzo[*b*][1,4]dioxine-2-carboxamide Limoniumin F 3,3′ -demethyl-heliotropamide Limoniumin A Limoniumin B Limoniumin C Limoniumin D 6-hydroxy-4-(4-hydroxy-3-methoxyphenyl)-2-(4-hydroxyphenethyl)-7-methoxy-1*H*-benzo(*f*)isoindole-1,3(2*H*)-dione Cannabisin I Limoniumin E Limoniumin G Limoniumin H Limoniumin I Cannabisin D Cannabisin B Cannabisin C Cannabisin A Cannabisin F Thoreliamide B	**(33)** **(34)** **(35)** **(36)** **(37)** **(38)** **(39)** **(40)** **(41) ** **(42)** **(43)** **(44)** **(45)** **(46)** **(47)** **(48)** **(49)** **(50)** **(51)**
		Phenolic amide	*N*-*cis*-feruloyl tyramine *N*-*trans*-feruloyl tyramine	

* Chemical structures of the isolated compounds investigated for their cytotoxicities, numbered in bold, were drawn in the Appendix A.

**Table 3 pharmaceuticals-16-00293-t003:** Results of inhibitory concentration of 50% (IC_50_) of cell proliferation based on eligible studies included in the systematic review.

Reference Number	Cell Line	IC_50_ (µg/mL)/Compound Tested	Selectivity Index (SI)/ Compound Tested
**[18]**	A549	29 (DCM extract), >200 (EtOH extract), 110 (MeOH extract), >200 (Hex extract), 10 (Etoposide PC)	NR
	DLD-1	85 (DCM extract), >200 (EtOH extract), >200 (MeOH extract), >200 (Hex extract), 80 (Etoposide PC)	
	WS1	>200 (DCM extract), >200 (EtOH extract), 140 (MeOH extract), >200 (Hex extract), 26 (Etoposide PC)	
**[33]**	HepG2	>200 (CE), 67.97 (Aq fraction), 59.47 (EAF)	0.48 (CE) 2.94 (AF) 1.27 (EAF)
	HL-60	61.69 (CE), 49.68 (Aq fraction), 17.26 (EAF)	1.56 (CE) 4.02 (AF) 4.39 (EAF)
	PBMC	96.78 (CE), >200 (Aq fraction), 75.82 (EAF)	NR
	T-47D	90.68 (CE), >200 (Aq fraction), 77.70 (EAF)	1.08 (CE) 1.00 (AF) 0.48 (EAF)
	HL-60	53.27 (SFa), 35.48 (SFb), 44.28 (SFc), 41.63 (SFd), 43.62 (SFe), 8.21 (SFf), 7.35 (SFg), 45.58 (SFh), 55.60 (SFi), 54.06 (SFj), 53.32 (SFk), 1.0 (Amsacrine PC)	NR
	K562	43.72 (SFa), 52.21 (SFb), 52.75 (SFc), 43.95 (SFd), 47.79 (SFe), 36.13 (SFf), 40.88 (SFg), 49.91 (SFh), 51.85 (SFi), 50.16 (SFj), 37.77 (SFk), 0.9 (Amsacrine PC)	
	MOLT-4	37.43 (SFa), 34.34 (SFb), 35.99 (SFc), 46.47 (SFd), 45.25 (SFe), 40.42 (SFf), 7.92 (SFg), 20.36 (SFh), 54.92 (SFi), 52.76 (SFj), 9.62 (SFk), NR (Amsacrine PC)	
	PANC-1	>100 (SFa), >100 (SFb), 76.81 (SFc), 45.54 (SFd), >100 (SFe), 58.65 (SFf), >100 (SFg), >100 (SFh), >100 (SFi), >100 (SFj), >100 (SFk), >100(Amsacrine PC)	
	SK-MEL-28	NA	
	Toledo	57.46 (SFa), 57.02 (SFb), 61.29 (SFc), 54.09 (SFd), 55.29 (SFe), 55.29 (SFf), 54.32 (SFg), 57.36(SFh), 60.65 (SFi), 58.68 (SFj), 59.38 (SFk), 0.5 (amsacrine PC)	
	K562	37.04 **(28)**, 29.24 **(29)**, 51.17 **(30)**	2.69 **(28)** 3.41 **(29)** 1.95 **(30)**
	Vero	>100 **(28)**, >100 **(29)**, >100 **(30)**	NR
**[37]**	HCT116	7.60 (BuOH extract), 25.74 ***** **(31)**, NA **(32)**	NR
**[22]**	MCF-7	19.65 and 14.57 (PE extracts), 17.60 and 21.8 (DCM extracts), 8.70 and 17.18 (MeOH extracts), 3.39 (Doxorubicin PC)	NR
	HepG2	9.97 and 16.97 (PE extracts), 20.62 and 11.15 (DCM extracts), 13.90 and 24.86 (MeOH extracts), 7.38 (Doxorubicin PC)	NR
**[39]**	HeLa	25.25 (EtOAc extract), NA (**(2)**, **(7)** and **(13)**), 19.24 ***** **(17)**, 12.85 * **(18)**, 31.57 * **(19)**, 0.23 * (Doxorubicin PC)	NR
	MCF-7	NA (EtOAc extract), 20.08 **(2)**, 21.58 **(7)**, 43.28 **(13)**, 28.85 **(17)**, 14.14 **(18)**, NA **(19)**	

Abbreviations: A549: human lung carcinoma; Aq: aqueous; BuOH: *n*-butanol; CE: crude extract; DCM: dichloromethane; DLD-1: human colorectal adenocarcinoma; EAF: ethyl-acetate fraction; HCT116: human colorectal carcinoma; HepG2: human hepatocellular carcinoma; EtOAc: ethyl-acetate; EtOH: ethanol; Hex: *n*-hexane; HL-60: human acute promyelocytic leukemia; MCF-7: human breast carcinoma; MeOH: methanol; MOLT-4: human acute lymphoblastic leukemia; NA: not affected; NR: not related; PANC-1: human pancreas epithelioid carcinoma; PBMCs: human primary peripheral blood mononuclear cells (normal cells); positive control (PC); PE: petroleum ether; RAW-264.7: Abelson murine leukemia virus-induced tumor (mouse); SFs: subfractions; SK-MEL-28: human malignant melanoma; T-47D: human ductal carcinoma; Toledo: human diffuse large cell lymphoma (non-Hodgkin’s B cell); Vero: kidney (monkey normal cell); WS1: human skin fibroblast (normal cell). ***** IC_50_ in µM. Note: Tested compounds were described in the order of crude extracts, fractions, subfractions, isolated compounds, and positive controls.

## Data Availability

Data sharing not applicable.

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
