# Peer review of "Chemical Constituents, Anticancer and Anti-Proliferative Potential of Limonium Species: A Systematic Review"

_pharmaceuticals, 2023, doi:10.3390/ph16020293_

Round 1
Reviewer 1 Report
This systemic review discussed the potential of Limonium spp as an anticancer source. Overall, the study was well-designed and presented. Some minors are suggested.
In table 2, align the numbers 1 to 10 to the compounds.
Are there any of these compounds that have been evaluated in clinical trials and how about their effects?
Authors may suggest future perspectives for the study in this field.
Author Response
Response to Reviewer 1 Comments
Point 1: In table 2, align the numbers 1 to 10 to the compounds.
Response 1: Thanks for your correction. The change in the table 2 was done as "track changes".
Point 2: Are there any of these compounds that have been evaluated in clinical trials and how about their effects?
Response 2: We carried out a quick survey on the subject and verified the existence of more than 3800 articles. Of these, around 200 articles would need to be read and check for possible inclusions or even exclusions. Thus, we understand the question, but it will not be possible to add precise information in the body of the manuscript.
Point 3: Authors may suggest future perspectives for the study in this field.
Response 3: About this question, a possible future perspective is related to the previous question. Furthermore, we are working with cell culture with Limonium brasiliense in the perspective of positive evaluation of colon and prostate cancer.

Reviewer 2 Report
Paper is well written

Author Response
Response to Reviewer 2 Comments
Point 1: Discussion section 4. Kindly check the numbering. 4.1, 2.2. Needs revising. Line 451. As per the guideline of various academic journals IMRaD format as an organizational structure to write the paper is acceptable i.e, Introduction, Methodology, Results and Discussion. However, if the editor requires to change the organizational structure, kindly correct the numbering.
Response 1: Thanks for your correction. The change in the numbering was done as "track changes". Line 460 and 473.
Point 2: Recheck the referencing numbers.
Response 2: We appreciate the contribution of the reviewer. The referencing numbers were checked and they are correct.
Point 3: Figures. Kindly see that the copyright permission has been taken for chemical structures from the authors of the publication.
Response 3: We appreciate the contribution of the reviewer. The chemical structures were drawn by ChemDraw version 14.0.0.118 (supplementary information, Appendix SF). We understand that chemical structure is not a drawing of something that has copyright, we understand that chemical structures are free knowledge from the moment they were published. Additionally, references related to chemical structures are duly cited in the text.
